# Surprise-Modulated Meta-Advantages in Reinforcement Learning: Towards Language-Neutral Post-Training for Code LLMs

## Abstract

Large language models are more beneficial for code generation in mainstream languages such as Python and JavaScript, however, they are very ineffective for resource-constrained languages such as Fortran, OCaml, and R. We rephrase this discrepancy not as a consequence of inevitable data lack of information, but as a problem in learning efficiency. In this work, we present PolyCode, which is trained by a groupwise meta-normalised Proximal Policy Optimization (PPO) which we refer to as GMPO. GMPO is a standard PPO-clip objective that has two new additions: (i) Cross-Group Meta-Normalization (CGMN) that suppresses variance by collecting meta-statistics across prompt similarities, and (ii) Surprise-Based Advantage Modulation (SBAM) that gives preference to updates where the reward signal deviates from a relative confidence of the model. We consequently enforce language neutrality of evaluation by input and output only by binary reward r in either 0 or 1 for exact conformity, and thus avoid the need for unit test translation across languages. Empirically, PolyCode-4B always matches or significantly exceeds smaller baselines on our Ag-LiveCodeBench-X benchmark with considerable improvements over WPLL for Fortran and OCaml. For a standardised reporting, pass@1 is defined as a Monte Carlo estimate derived from multiple single-sample trials (single draw 20 times per prompt reactance at T=0.2), but the best of selection and voting were not used during implementation.

## 1 Introduction

Large language model (LLM) has greatly changed the way we develop software applications; however, the advantages of large language models are not equally distributed across programming languages. Practitioners working within scientific and engineering ecosystems (i.e., Fortran, R, Julia, OCaml, and Lua) are faced with constraints both in the amount of data they use for training and signed, mature tooling. The Stack_V2 Lozhkov et al. (2024a) shows that there are strong disparities and increase a self-affirming cycle whereby - low resource languages are at the bottom in terms of model quality and robustness of evaluation infrastructures. We propose PolyCode, which is trained by a GMPO-type PPO-like policy gradient approach enhanced with a so-called cross-task meta normalized (CGMN) and a surprise-based advantage modulation (SBAM). CGMN alleviates the variance of a batch of local statistics over *similar* prompts to minimise variance in regimes with low signal; SBAM approaches the regimen of the samples in which the observed reward is contrary to the model's *meta-not normalised sequence likelihood* (relative confidence). This approach is coupled to the language-neutral IO - only execution harness that can evaluate programmes only in terms of deterministic stdin / stdout behaviour so as to eliminate the necessity for per language unit test translations.

**Scope and non-claims.** We put interviewer-level interventions on top of PPO-clipped. We do not add datasets and decoding tricks, as well as proprietary unit-test graders. Where we use Ag-LiveCodeBench-X (evaluation) and Ag-Codeforces-X (training) these are split, reconstruction-based splits taken from publicly available sources and have not been created with any new samples; which we use to ensure the I/O-only protocol is consistent and auditable. Our aim is to isolate algorithmic effects (Figure 2) while holding infrastructure and decoding fixed.

On Ag-LiveCodeBench-X (from LiveCodeBench Jain et al. (2024)) at MultiPL-E Cassano et al. (2023), PolyCode-4B provides significant advantages under lower-resources across languages maintaining the competitiveness across better-resourced ones under a single draw pass@1 protocol. All assertions are with respect to this conservative environment: no best-of, no majority voting, single-point templates, and capture of decontamination and run-time cheques.

**Design philosophy.** The most significant challenges are not based solely on lack of data, but stem from the gap between surface representations based on language as opposed to computationally invariant representations based on the task. In both sparse and noisy models, such normalisation on individual prompts may suffer from high variance whereas indiscriminate accumulation over heterogeneous prompts could result in bias into the updates. CGMN reduces this through calculating the neighbourhood-weighted meta-statistics, which utilises the local structure in an attempt to stabilise scale without overlooking local structural dependencies. SBAM complements this by sign-preserving rescaling when reward disagrees with *relative* confidence, turning confident failures and hesitant successes into disproportionately informative updates, all while *preserving* the PPO-clip geometry.

**Clarifying relation to GRPO and PPO.** We leverage GRPO inspired grouped sampling to provide robust within prompt statistics to augment a PPO clip objective aimed at the advantage scale-which is numerically accomplished using meta normalisation and surprise modulation instead of asymptotic relative ranking surrogate. This way, GMPO is still fully compatible with conventional PPO theoretical frameworks and implementation infrastructures, which allows an easy integration into existing reinforcement-learning pipelines.

**Contributions.**

- **GMPO**, groupwise meta-normalised Proximal Policy Optimization (GP-MPO), carries a meticulously extended greediness sophistication, referred to because surprise modulation. Its clustered sampling interface is separated from the PPO objective, so it directly/decentralised controls variance and information content through its advantage scaling.

- **Language-neutral execution**, lets per-language engineering is done to a bare minimum of manifest specs, while remaining purely behavioural (stdin/stdout) while being portable across programming languages.

- **Empirical gains** across Fortran, Julia, Lua, OCaml, and R using multiple model families Guo et al. (2024); Microsoft et al. (2025), under identical decoding and infrastructure.

## 2 BACKGROUND AND RELATED WORK

Code-oriented pretraining improves general reasoning Ma et al. (2023) and can be realized via code-only training Lozhkov et al. (2024b); Gehring et al. (2025) or continued pretraining from general LMs Rozière et al. (2024). However, the ecosystem remains skewed toward high-resource languages Lozhkov et al. (2024b); Athiwaratkun et al. (2023); Wang et al. (2023). Reinforcement learning (RL) for code advances beyond supervised fine-tuning Wang et al. (2025), with execution feedback Gehring et al. (2025), prolonged RL Hu et al. (2025), and rule-based rewards DeepSeek-AI et al. (2025). RL pipelines, however, often depend on language-specific infrastructure and extensive unit-test harnesses, which are less available for low-resource languages. Our method targets this gap by (i) avoiding per-language test translation, (ii) leveraging cross-task structure to stabilize learning in low-resource settings, and (iii) clarifying multi-language evaluation practices. We situate our method within groupwise sampling settings (as in GRPO Shao et al. (2024)) while keeping optimization strictly *PPO-clip*, with meta-normalization and surprise modulation layered on top.

## 3 PRELIMINARIES AND NOTATION

We consider program-synthesis tasks indexed by prompts $x \in \mathcal{X}$, where outputs $y$ are complete programs emitted in one shot by a policy $\pi_\theta(y|x)$. Programs are compiled and executed in a sandbox; rewards are **binary** and deterministic:

$r(x, y) \in \{0, 1\}$ with $r(x, y) = 1$ iff I/O matches exactly (format, precision, delimiters).

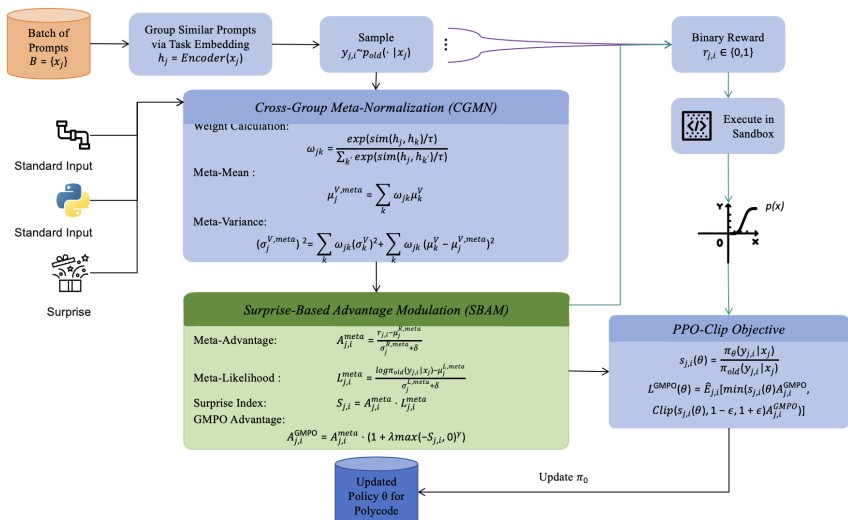

Figure 1: GMPO Training Pipeline for PolyCode.

We use the *sequence-level* log-likelihood $L(x, y) = \log \pi_\theta(y|x)$ when appropriate. Unless noted, expectations $\mathbb{E}$ are taken over data batches and sampling randomness. The *PPO clipping radius* is $\varepsilon \in (0, 1)$; *numerical stabilizers* used in denominators are denoted by $\delta > 0$.

## 4 METHOD

### 4.1 PROBLEM SETUP AND BEHAVIORAL TASKS

We restrict to deterministic programs reading from `stdin` and writing to `stdout`. Each prompt carries canonical input-output examples and format rules. The reward is *binary*: exact conformance yields $r=1$, otherwise $r=0$ (compilation or runtime failures also map to 0). This abstraction makes the benchmarking language-agnostic and as a result, cross-language benchmarking easy. Though binary rewards are sparse, the complement of variance reduction provided by CGMN and concentration of benefits provided by SBAM makes PPO-styled updates viable without the need to do fiddly reward-shaping. The obtained underlying structure is the same on all programming languages; therefore, observed differences in performance should be explained by policy behaviour rather than peculiarities in the evaluator.

### 4.2 GMPO: GROUPWISE META-NORMALIZED PPO

GMPO uses *grouped sampling per prompt*: for each $x_j$ in batch $\mathcal{B}$, draw $G$ responses $\{y_{j,i}\}_{i=1}^G$ from $\pi_{\text{old}}(\cdot|x_j)$. Grouping supports (i) per-prompt statistics and (ii) cross-task meta-normalization; optimization remains *PPO-clip*.

**Cross-Group Meta-Normalization (CGMN).** For each $x_j$, compute a task embedding $h_j = \text{Encoder}(\pi_{\text{old}}, x_j)$ (no gradient). Define batch-local softmax weights

$$w_{jk} = \frac{\exp(\text{sim}(h_j, h_k)/\tau)}{\sum_{k'} \exp(\text{sim}(h_j, h_{k'})/\tau)}, \qquad w_{jk} \geq 0, \ \sum_k w_{jk} = 1. \tag{1}$$

For $V \in \{R, L\}$, let $\mu_k^V$ and $(\sigma_k^V)^2$ be per-prompt sample statistics over $\{y_{k,i}\}_{i=1}^G$. Batch-local meta-statistics follow the law of total variance:

$$\mu_j^{V,\text{meta}} = \sum_k w_{jk} \, \mu_k^V, \tag{2}$$

$$(\sigma_j^{V,\text{meta}})^2 = \sum_k w_{jk} \, (\sigma_k^V)^2 + \sum_k w_{jk} \, (\mu_k^V - \mu_j^{V,\text{meta}})^2. \tag{3}$$

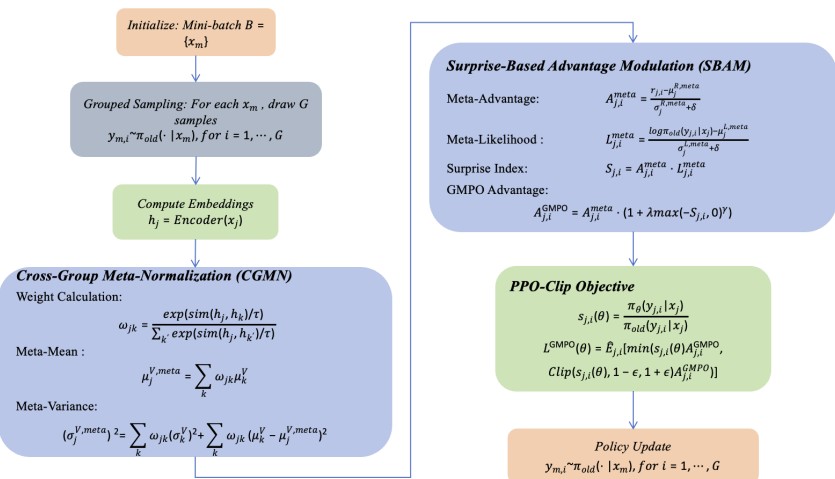

Figure 2: Algorithm Flowchart.

A critic-free normalized advantage proxy is

$$A_{j,i}^{\mathrm{meta}} = \frac{r(x_j, y_{j,i}) - \mu_j^{R,\mathrm{meta}}}{\sigma_j^{R,\mathrm{meta}} + \delta}, \quad \delta > 0. \tag{4}$$

Owing to the pooling of statistical strength on neighbouring prompts, CGMN yields a decrease in variance of Ameta under a low resource scenario. For the asymptotic case in which the weighting coefficients wk get more and more concentrated on k=j, CGMN approaches the standard single-difference confirmatory per prompt standardisation. Alternatively, when early similarities vanish so much that prompt similarities can't be observed anymore CGMN falls back to batch-wise normalisation which normalises the scale without introducing a substantial controlled bias.

**Sequence Likelihood Normalization.** Let $L_{j,i} = \log \pi_{\mathrm{old}}(y_{j,i}|x_j)$. Define

$$\widehat{L}_{j,i}^{\mathrm{meta}} = \frac{L_{j,i} - \mu_j^{L,\mathrm{meta}}}{\sigma_j^{L,\mathrm{meta}} + \delta}. \tag{5}$$

This places *relative* confidence on a comparable scale across prompts. Meta-normalization mitigates, but does not entirely remove, known length effects of sequence-level likelihood; we therefore accompany results with diagnostics (Section 8) and note token-level variants in Appendix K.

**Surprise-Based Advantage Modulation (SBAM).** Define $S_{j,i} = A_{j,i}^{\mathrm{meta}} \cdot \widehat{L}_{j,i}^{\mathrm{meta}}$ and modulate

$$A_{j,i}^{\mathrm{GMPO}} = A_{j,i}^{\mathrm{meta}} \cdot \left(1 + \lambda \, \phi(-S_{j,i})\right), \qquad \lambda > 0, \tag{6}$$

with the stable ramp

$$\phi(u) = \left[\max(u, 0)\right]^{\gamma}, \qquad \boxed{\gamma = 1}. \tag{7}$$

Hence, *linear* amplification applies when $S_{j,i} < 0$ (confident failures or hesitant successes under *meta-normalized* likelihood) and no amplification otherwise.[1] SBAM *preserves the sign* of the advantage and scales it monotonically in a function $\phi$ that is *globally 1-Lipschitz* (but merely subdifferentiable at 0), avoiding gradient blow-ups near $S=0$.

---

[1] When $0<\gamma<1$, derivatives near $S \to 0^-$ can become large; we adopt $\gamma = 1$ for disciplined behavior. Smooth bounded ramps (e.g., softplus) are drop-in alternatives with similar qualitative effects.

---

**Algorithm 1** GMPO Training (Figure 1)

---

1: Sample mini-batch $\mathcal{B} = \{x_j\}$; for each $x_j$ draw $G$ samples $y_{j,i} \sim \pi_{\text{old}}(\cdot|x_j)$
2: Execute each $(x_j, y_{j,i})$ in sandbox; collect binary rewards $r_{j,i} \in \{0, 1\}$
3: Compute task embeddings $h_j$ (no grad); compute $w_{jk}$ within batch
4: Compute per-prompt stats $\mu_j^V, \sigma_j^V$ and meta-stats $\mu_j^{V,\text{meta}}, \sigma_j^{V,\text{meta}}$ for $V \in \{R, L\}$
5: Form $A_{j,i}^{\text{meta}}, \widehat{L}_{j,i}^{\text{meta}}, S_{j,i}$, and $A_{j,i}^{\text{GMPO}}$
6: Update $\theta$ by ascending PPO-clip surrogate with KL regularization
7: Set $\pi_{\text{old}} \leftarrow \pi_\theta$ periodically

---

**PPO-Type Objective.** Let $s_{j,i}(\theta) = \frac{\pi_\theta(y_{j,i}|x_j)}{\pi_{\text{old}}(y_{j,i}|x_j)}$. GMPO maximizes

$$\mathcal{L}^{\text{GMPO}}(\theta) = \mathbb{E}_{\mathcal{B}} \left[ \frac{1}{G} \sum_{j,i} \min\left( s_{j,i}(\theta) A_{j,i}^{\text{GMPO}}, \ \text{clip}\left(s_{j,i}(\theta), 1-\varepsilon, 1+\varepsilon\right) A_{j,i}^{\text{GMPO}} \right) \right]$$
$$- \ \beta \, \text{KL}(\pi_\theta \,\|\, \pi_{\text{ref}}) \quad (8)$$

with PPO clipping parameter $\varepsilon$ and optional KL penalty $\beta \geq 0$ (commonly to $\pi_{\text{old}}$). This makes explicit that GMPO is PPO-type with grouped sampling and meta-normalized, surprise-modulated advantages.

### 4.3 Design Choices, Reductions, and Edge Cases

**Batch-local neighborhoods.** Batch-local $w_{jk}$ keeps compute predictable and avoids memory banks while providing sufficient coverage in typical batch sizes. A memory-bank variant is compatible (Appendix J) but not required.

**Sequence vs token granularity.** We use sequence-level ratios and likelihoods for simplicity and coupling to sequence-defined binary rewards. Token-level variants (Appendix K) are compatible and reduce residual length effects; in practice, sequence-level normalization plus CGMN already stabilizes scales across prompts.

**Edge cases.** When $G=1$, per-prompt statistics degenerate; CGMN still aggregates across prompts and remains useful. When similarities collapse (nearly uniform $w_{jk}$), meta-statistics default to batch-wide normalization, stabilizing scale with a small bias that is attenuated by top-$K$ neighborhoods (Appendix L).

**Modulation family.** The linear ramp $\gamma=1$ ensures disciplined, sign-preserving rescaling with global 1-Lipschitz continuity (subdifferentiable at 0). Bounded smooth alternatives limit growth under extreme surprises without altering qualitative behavior (Appendix M).

### 4.4 Complexity and Memory Considerations

Computing all $w_{jk}$ is $O(|\mathcal{B}|^2)$; top-$K$ truncation yields $O(|\mathcal{B}|K)$. Per-batch statistics cost $O(|\mathcal{B}|G)$. Memory scales with buffered logits for sequence likelihoods; sequence-level quantities keep this modest relative to token-level variants. In distributed settings, only neighborhood summaries (means, variances) require cross-replica reduction.

### 4.5 Algorithmic Outline

Algorithm 1 summarizes one epoch. The encoder for $h_j$ is detached to avoid coupling representation learning to transient batch composition. We periodically update $\pi_{\text{old}}$ and optionally anneal $\beta$.

### 4.6 Language-Neutral Runtime and Minimal Configuration

We extract programs from model outputs, compile and execute them inside OCI containers with limits on CPU, memory, wall-clock, and stdout size. Each language is registered via a minimal YAML

manifest specifying installation, compile, and run commands (Appendix D). Installation occurs at image build time; evaluation runs offline without network access. We report orchestration overheads at the scheduler layer qualitatively to contextualize runtime invariants; these do not change our conclusions.

## 5    DATASETS AND DECONTAMINATION

**Training.**    **Ag-Codeforces-X** is a reconstruction-oriented split derived from Open-R1 Codeforces Penedo et al. (2025), keeping I/O task format intact. We also construct an MBPP-based variant to probe transfer from elementary problems to harder evaluation, following common practice that HumanEval-style tasks Chen et al. (2021) are easier than competition-style programs. No new samples are introduced beyond upstream sources.

**Evaluation.**    We use (i) **MultiPL-E** Cassano et al. (2023) for cross-lingual function-style evaluation and (ii) **Ag-LiveCodeBench-X**, adapted from LiveCodeBench Jain et al. (2024), for competition-style I/O tasks. All LiveCodeBench-derived problems are excluded from training through a two-sided screen.

**Decontamination Protocol.**    We apply canonicalization, exact hashing, near-duplicate screening using $n$-gram MinHash/LSH for both text and code, AST shingling where parsers exist, random sampling for manual review near thresholds, and logging of exclusion lists (IDs and hash digests). We also run our screen on third-party datasets that claim decontamination (Appendix B). The goal is auditable exclusion of overlaps without altering benchmark content.

## 6    EVALUATION PROTOCOL AND STATISTICAL REPORTING

**Unified Decoding.**    We use per-language prompt templates with uniform decoding: temperature $T{=}0.2$ and sufficient max length to avoid truncation; *no best-of reranking* or *majority voting* for primary pass@1 numbers. Templates are listed in Appendix C and kept fixed across models.

**Pass@1 Estimation (Monte Carlo, No Best-of).**    For each prompt, take $M{=}20$ independent single-sample draws at $T{=}0.2$, evaluate each once, and estimate pass@1 as $\hat{p} = \frac{1}{M} \sum_{m=1}^{M} \mathbf{1}[\text{success}]$. This estimates the *single-draw* success probability under the stated decoding distribution; it performs no reranking or voting. Uncertainty reporting (e.g., Wilson intervals) and macro-averaging over prompts are detailed in Appendix A. All percentages should be interpreted alongside the number of evaluated prompts and the corresponding uncertainty envelopes.

**Consistency and Auditability.**    We log compilation/execution outcomes, seeds, and decoded outputs per prompt. Decoding templates and sandbox manifests are fixed across models. When a compile fails, reward is 0; we retain stderr/stdout to diagnose failure modes (Appendix H).

## 7    EXPERIMENTS

### 7.1    SETUP AND CONTROLS

All of the models considered in this study have the same decoding budget and follow the language templates defined in Section 6. The results reported are point constructions of pass@1 based on the basis of the MC estimator described above. Timestamp, random seed and testing results are recorded for auditability purposes. Training schedule, batch organisation and container image are also fixed across all experimental variants, to only allow purely algorithmic effect characterisation with toggles toggled on or off.

**Training protocol and data mixing.** Each reinforcement learning run consists of lightweight on-policy sampling and off-policy evaluation runs running back-to-back inside one single single containerized reinforcement learning harness. Proximal Policy Optimization clipping is used in conjunction with the KL divergence to the reference policy which is updated periodically to decrease

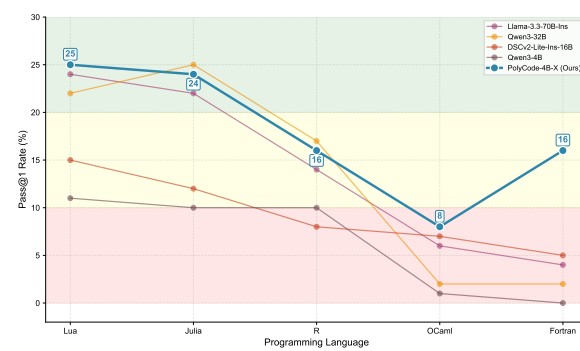

Figure 3: Performance comparison of PolyCode-4B-X models across five low-resource languages on Ag-LiveCodeBench-X benchmark. GMPO-trained 4B parameter models have competitive results against much larger baselines (16B–70B parameters), supporting the effectiveness of meta-normalization and surprise modulation.

the KL divergence at all times. Mini-batches are adjusted so that prompts of similar structural types such as string parsing, numeric formatting, and combinatorial are clustered together enough to fill up local neighborhoods for CGMN but with enough variety so that degenerate structures are avoided. The reward function is an I/O-only one, this means that the amplitudes of the signals stay the same for all languages.

**Infrastructure parity and logging.** We pin container images and toolchain versions, disable network access during execution, and record compile commands, exit codes, and the first bytes of stdout/stderr for each attempt. Seeds of sampling and data loader shuffling are recorded along with the success indicators at the prompt level. This allows the reproducibility of any prompt-draw pair thereby allowing the pass@1 calculations to be independently verified using these artefacts.

## 7.2 PRIMARY RESULTS AND DYNAMICS

Figure 3 summarizes Ag-LiveCodeBench-X results: the baseline Qwen3-4B achieves $11\%$ pass@1 on Lua and $0\%$ on Fortran; after GMPO training, PolyCode-4B-X reaches $25\%$ on Lua and $16\%$ on Fortran, competitive with or better than larger baselines. We attribute variance reduction from the use of CGMN through cross-task similarity and the emphasis on informative errors of SBAM to languages with low available resources marked by reward sparsity and miscalibration of relative confidence. As mentioned already, the given percentages are one-sample probabilities based on the fixed decoding distribution and are not best-of metrics.

**Learning dynamics.** In the early stages of the training, the updates mostly prevent surface level failures (e.g. syntax errors, lacking import statement, off-by-one format discrepancy etc.) because such errors occur with high certainty level and thus are enhanced by according SBAM itself. And the percent of refurbishing errors in the format of the compilation goes upward and adhere semantic error rate to a corresponding number of errors of the boundary conditions.

**Case narratives from under-served languages.** In the Fortran language, a popular type of errors is related to the usage of scientific notation and width specifiers. Programmes that contain no errors (compiled successfully) have more often than not, generated outputs with leading spaces or incorrectly signed exponent. SBAM magnifies the gradient impact of these high-calibre failures and so encourages edits to FORMAT statements or explicit WRITEs as opposed to the fundamental logic. Like OCaml, we do have a lot of failures that stem from forgotten open's or incomplete pattern match, and the action of giving priority with confident errors will attract the interest less on the missing modules and more on logic for boundary conditions.

## 7.3 GENERALIZATION TO FUNCTION-STYLE EVALUATION

Despite training on I/O-style tasks, function-mode prompting elicits function-conformant solutions on MultiPL-E (Figure 4). This suggests that the learned improvements target algorithmic com-

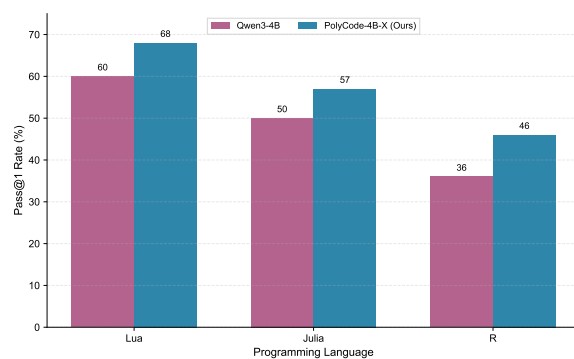

Figure 4: Generalization on MultiPL-E. Improvements across Lua, Julia, and R indicate that training on I/O tasks transfers to function-style unit-test evaluation when guided by simple function-mode templates.

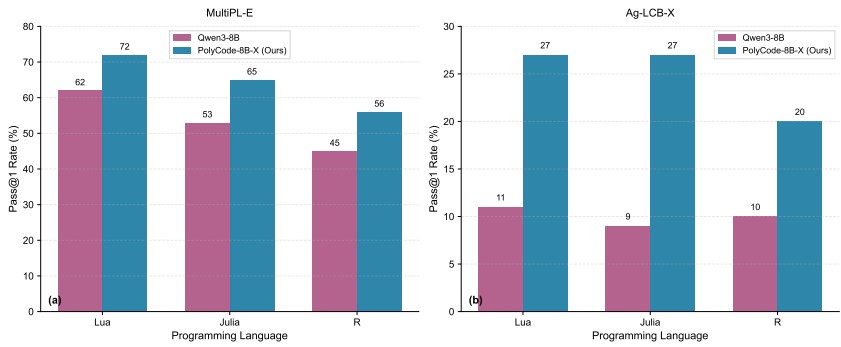

Figure 5: Scaling behavior with 8B models. The meta-normalization framework preserves efficacy at higher capacity, showing compounding benefits on both benchmarks.

petence rather than I/O-specific patterns: reductions in surface/format errors carry over as fewer extraneous prints, cleaner return values, and more deliberate guard conditions.

### 7.4 DIAGNOSTICS AND FAILURE TAXONOMY

We use a four-fold taxonomy consisting of: (i) occurrences of errors at a surface level (e.g. syntactic violations, missing imports, non-existent APIs); (ii) problems in format coverage (i.e. I/O layout, precision, delimiters); (iii) semantic inconsistencies (i.e. algorithmic boundary conditions); and (iv) performance failures (e.g., timeouts). Our GMPO approach significantly reduces the first two categories especially in under-represented languages and the diagnostic labels are facilitating qualitative analysis, but are not included in the quantitative.

## 8 THREATS TO VALIDITY AND PRACTICAL CONSIDERATIONS

**Internal validity.** Template drift and unconscious best-of selection are the main sources of confounding which are controlled for by fixing the templates as well as including the exclusion of best-of set wtv or voting scheme from the main metric. Imposing Pass@1 rather than Pass@k offers a focus on base capability; while other figures such as Pass@$k$ may give more positive outcomes, they are squarely out of the scope of the current study. However, although we have used five different languages (two different flavours of evaluation style) our results do not necessarily generalise in GUI/HTML/HTML-based service contexts or logic/array involve-oriented paradigms.

**Residual length effects.** Sequence level likelihood is correlated with programme duration. Notwithstanding that meta-normalisation reduces the length-span effect, some effects of the residual length on text quality may still hold, particularly between languages that display different stylistic pat-

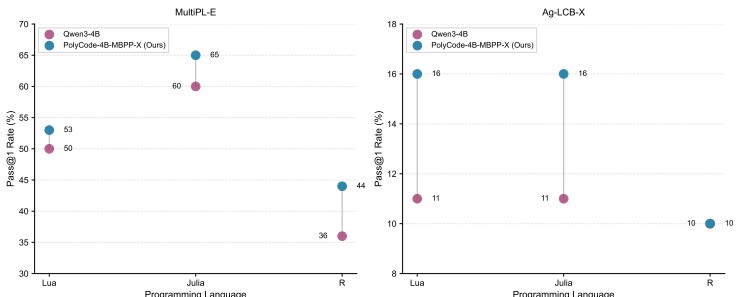

Figure 6: Training on simpler MBPP yields smaller but tangible gains, suggesting that SBAM benefits from training distributions with sufficient difficulty and diverse failure modes.

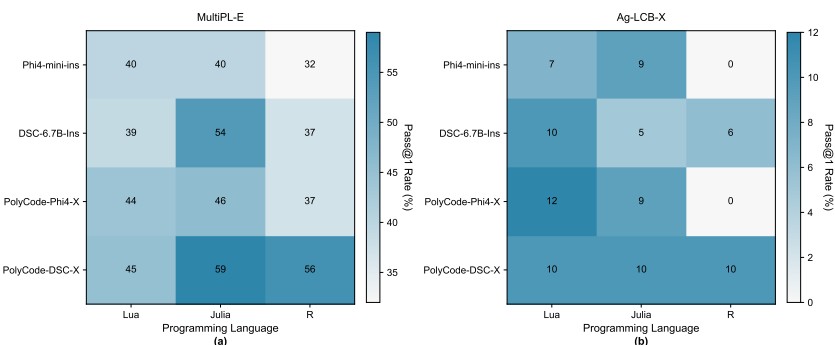

Figure 7: Architecture-neutrality: applying the same GMPO recipe to alternative families shows consistent improvements without language-specific test translation.

terns. A further model justification of performance improvements over achieved results in related studies confirms that the observed improvements over existing work rates are not solely a matter of programme length of n, as was classed by other workers (cf. Appendix K for comparisons on a token-level).

## 9 CONCLUSION

We propose PolyCode and GMPO showing that cross task meta normalisation combined with surprise-based attention which is applied in a PPO-type objective enhances the multilingual coding competence of compact models within resource constrained scenarios. By standardising evaluation protocols and placing less voltage on language specific engineering, we move towards the goal of equal-footing AI code assistance featuring computational structure.

## 10 REPRODUCIBILITY STATEMENT

As shown in Appendix P.

## 11 ETHICS STATEMENT

As code generation involves sensitive contexts in security applications, the use of a sandboxed execution environment and of resource limits are made available to mitigate the risk associated with training and evaluation.

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

## A  STATISTICAL ESTIMATION DETAILS

**Per-Prompt Estimator.**  For each prompt, run $M$ i.i.d. single-sample draws, obtain $\hat{p} = \frac{1}{M}\sum_{m=1}^{M} \mathbf{1}[\text{success}]$, and report Wilson intervals with center and halfwidth:

$$\hat{p}_W = \frac{\hat{p} + \frac{z^2}{2M}}{1 + \frac{z^2}{M}}, \qquad \text{halfwidth} = \frac{z\sqrt{\frac{\hat{p}(1-\hat{p})}{M} + \frac{z^2}{4M^2}}}{1 + \frac{z^2}{M}},$$

where $z = z_{\alpha/2}$. All primary numbers refer to single-draw pass@1 estimates *without* best-of and *without* voting.

**Macro vs. micro averaging.**  Macro-averaging is the calculation of the unweighted average over the prompts, micro-averaging assigns the weights to the prompts based on the frequency of their occurrence in the corpus and can over-emphasize categories that contain more items. For cross-language parity, we are using macro-averaging.

**Seed discipline and independence.**  Separate seeds they are used for data-loader shuffling and sampling randomness. To minimize the correlation between cross-prompt drawing of samples, we sample decoding randomness for each prompt-draw pair. When reporting the means from seeds averaged over, we are reporting mean $\pm$.

**Bootstrap and delta methods.**  Nonparametric bootstrap over prompts for variability of macro-averaged pass@1 Delta-method approximations for delta-intervals under the weak dependence.

**Length-related diagnostics.**  Because length affects the likelihoods at the sequence level, we test for the difference in the distribution of values of the estimator of specificity of meta-haplotypes, the likelihoods, of successes vs. failures, to ensure that improvements are not artifacts of length. Token level diagnostics with a function of robustness cheque (Appendix K).

## B  DECONTAMINATION PSEUDOCODE, INVARIANTS, AND AUDIT TRAIL

---
**Algorithm 2** Two-Sided Decontamination

---
1: **Input:** Train set $\mathcal{T}$ (prompts, solutions), Eval set $\mathcal{E}$
2: Canonicalize: lowercase, trim spaces, strip comments/headers; normalize numeric literals and whitespace
3: Hash exact strings and remove duplicates (prompt text and canonical code)
4: Build $n$-gram MinHash/LSH indices for prompts and code (both $\mathcal{T}, \mathcal{E}$)
5: Remove pairs with Jaccard similarity $\geq \theta_{\text{text}}$ or $\geq \theta_{\text{code}}$ (thresholds chosen conservatively)
6: **if** parsers available **then**
7:   Parse code to AST, shingle subtrees, MinHash/LSH; remove near-duplicates by structure
8: **end if**
9: Sample borderline clusters near thresholds; manual review and prune if necessary
10: Log and export exclusion list (IDs, hash digests) for auditability

---

**Invariants.**  (1) No token-level or AST-level structure from $\mathcal{E}$ appears in $\mathcal{T}$ after screening; (2) Canonicalization is idempotent and language-agnostic; (3) Screening is re-run whenever $\mathcal{T}$ or $\mathcal{E}$ changes; (4) The exclusion list is stable under re-hashing.

**Borderline cluster adjudication.**  For clusters near thresholds, we prefer conservative pruning. Reviewers adjudicate whether overlap is incidental (e.g., boilerplate) or substantive (algorithmic core), erring on the side of exclusion in ambiguous cases.

## C  PROMPTING TEMPLATES AND INTERFACES

### C.1  I/O MODE (TRAINING AND AG-LIVECODEBENCH-X)

```
You are a helpful assistant writing a complete PROGRAM in <LANG>.
Constraints:
- Read all input from STDIN.
- Write output to STDOUT.
- Deterministic behavior only; no network or file I/O.
- Follow exactly the specified format (spacing, newlines, precision).

<Problem statement and I/O specification here>

Now output ONLY the code. Do not include explanations or tests.
```

## C.2 FUNCTION MODE (MULTIPL-E)

```
You are writing a single function in <LANG> with the following signature:

<FUNCTION SIGNATURE HERE>

Implement ONLY this function. Do not add main(), I/O, prints, or tests.
Do not import nonstandard libraries unless stated.
```

We only add minimal language-specific boilerplate when required by the evaluator.

## D MINIMAL LANGUAGE MANIFESTS (YAML)

### D.1 FORTRAN (GFORTRAN)

```
language: fortran
install:
  - apt-get update
  - apt-get install -y gfortran
compile:
  - ["bash","-lc","gfortran Main.f90 -O2 -o Main"]
run:
  - ["bash","-lc","./Main"]
file_ext: ".f90"
stdin: true
stdout: true
```

### D.2 OCAML (OCAMLOPT)

```
language: ocaml
install:
  - apt-get update
  - apt-get install -y ocaml
compile:
  - ["bash","-lc","ocamlopt -o Main Main.ml"]
run:
  - ["bash","-lc","./Main"]
file_ext: ".ml"
stdin: true
stdout: true
```

## E IMPLEMENTATION NOTES

**Task embeddings and similarities.** We compute $h_j$ from a detached encoder representation; cosine similarity with temperature $\tau$ yields $w_{jk}$. Top-$K$ neighborhoods (Appendix L) reduce quadratic cost while preserving local structure.

**Distributed training.** Distributed training causes local computation of per-prompt statistics and meta statistics that are aggregated across the replicas using arithmetic means and the second moments. Only the aggregate statistics are reported from replica to replica, thus the communication overhead is small.

**Regularization and trust.** Additionally, a Kullback-Leibler loss penalty with respect to the previous policy p old is optionally introduced for the purposes of policy update stabilisation. Entropy set bonuses are orthogonal to this term. Please note that while it is possible to use stronger KL constraints or trust-region formulations to bound the importance ratios s, our analysis uses the clipped surrogate objective used in Proximal Policy Optimization, in which case it gives disciplined directionally strong guarantees.

**Gradient hygiene.** To minimise numerical instability, all z-scores are propagated with numerically stable gradient clipping/addition of denominator stabilisers d. Thresholding may be done on the extreme z-scores associated with he words with no impact to the UL of meta words of the same root or metawordyond practical ordering of the words' updates.

**Logging and debuggability.** The logging infrastructure is used to record the per prompt meta statistics, success/failure flags, and error categories. Standard error and Out are saved for examination and the non-ending runs are stopped by wall timeouts.

# F  THEORETICAL REMARKS ON STABILITY

**Approximate centering of meta-normalized advantages.** Let $Z_{j,i} = (r_{j,i} - \mu_j^{R,\text{meta}})/(\sigma_j^{R,\text{meta}} + \delta)$ with $\delta > 0$. For fixed meta-statistics,

$$\mathbb{E}_i[Z_{j,i}] = \frac{\mu_j^R - \mu_j^{R,\text{meta}}}{\sigma_j^{R,\text{meta}} + \delta},$$

which is generally nonzero unless $\mu_j^{R,\text{meta}} \approx \mu_j^R$. With neighborhood weights $w_{jk}$ concentrating on prompts similar to $x_j$, the bias term is typically small; normalization stabilizes scale and reduces sensitivity to reward sparsity.

**Proposition 1** (Range of the PPO-clip surrogate). *Let $A \in \mathbb{R}$ and $s = \frac{\pi_\theta(y|x)}{\pi_{old}(y|x)}$. Define $f(s) = \min(sA, \text{clip}(s, 1-\varepsilon, 1+\varepsilon) A)$ with $\varepsilon \in (0,1)$. Then*

$$A > 0: \quad f(s) \le (1+\varepsilon) A; \qquad A < 0: \quad f(s) \le (1-\varepsilon) A.$$

*Moreover, for $A < 0$ no uniform lower bound exists, since $s \to \infty$ implies $f(s) \to -\infty$.*

**Monotonicity of SBAM rescaling.** For fixed $A^{\text{meta}}$, $A^{\text{GMPO}}$ increases monotonically with the nonnegative ramp $\phi(-S)$ and preserves the sign of $A^{\text{meta}}$. With $\gamma=1$, $\phi$ is *globally 1-Lipschitz* (subdifferentiable at 0), which avoids gradient instabilities near $S = 0$.

**Reductions.** If $w_{jk} = \mathbf{1}[j = k]$ and $\lambda = 0$, GMPO reduces to PPO with per-prompt standardization. If $G = 1$ and neighborhoods are uniform, GMPO reduces to PPO with batch-wise standardization, still stabilizing scale.

**Neighborhood bias.** Because CGMN aggregates from a soft neighborhood, a controllable bias arises whenever neighboring prompts differ in reward difficulty. Temperature $\tau$ and top-$K$ truncation bound this effect; empirically, similarities derived from the detached encoder track reward scale sufficiently well to keep the bias modest.

# G  WORKED EXAMPLE OF CGMN

Consider prompts $x_1, x_2, x_3$ with per-prompt reward stats $(\mu_k^R, \sigma_k^R)$ and cosine similarities forming $w_{jk}$. The meta-mean $\mu_j^{R,\text{meta}}$ is $\sum_k w_{jk}\mu_k^R$, while meta-variance adds within-prompt variances and

between-prompt dispersion $\sum_k w_{jk}(\mu_k^R - \mu_j^{R,\text{meta}})^2$. When $x_1$ is low-resource with noisy estimates, contributions from $x_2, x_3$ stabilize scaling even when $G$ is small.

## H   Sandbox Execution Harness

---
**Algorithm 3** Sandboxed Execution and Rewarding

---
 1: Extract code block for target language; write to file with correct extension
 2: Compile using language manifest; if compile fails or times out, return reward 0
 3: Run binary in container with CPU/memory/time/output caps; feed canonical inputs
 4: If outputs match exactly (format, precision, delimiters), assign reward 1; else 0
 5: Log stderr/stdout and resource usage for analysis

---

**Determinism.**   We pin CPU/memory quotas and timeouts; environments are normalized to avoid locale/rounding differences that could affect formatting. Randomized hashing in certain runtimes is disabled where relevant.

## I   Ablation Design Grid

We recommend toggles: CGMN on/off; SBAM on/off; $\lambda$ sweeps with $\gamma=1$; batch-local vs memory-bank statistics; sequence-level vs token-level KL ($\beta \geq 0$); top-$K$ neighbor sizes. Each toggle isolates the effect of a single component under fixed decoding and runtime.

## J   Memory Bank Variant

A FIFO memory of recent $h_j$ vectors enables neighborhood computation beyond the current batch. The combination of coverage and staleness on neighbouring batch nodes is used to limit the memory size and the weights of nodes are renormalized over the union of memory and batch neighbours. Furthermore, summary statistics only are kept.

## K   Token-Level Variant

Seq-level variants are replaced with token-level variants in which the sum of log-probabilities in the token sequence (s) replaces likelihood, and at least moderate residual length effect can be eliminated. For clarity, simplicity, and to ensure a close relationship with the vulnerability-specific analysis of the binary rewards rewarded by the protocols, we rebuild the sequence-level variants; doing so requires extra bookkeeping and communication in order to aggregate per-account statistics per token.

## L   Top-$K$ Neighborhood Heuristic

We cap neighbors at the top-$K$ most similar prompts per $x_j$, normalizing $w_{jk}$ over this set. Complexity becomes $O(|\mathcal{B}|K)$. Choosing $K$ to exceed a cumulative weight threshold under the current $\tau$ yields robust neighborhoods without excessive compute.

## M   Alternative Modulation Families

Smooth bounded ramps such as $\mathrm{softplus}$ or $\tanh(u_+)$ ($u_+ = \max(u, 0)$) are drop-in; they limit growth for extreme surprises while retaining focus on $S<0$. We keep linear modulation as a principled, simple default.

## N  FAILURE TAXONOMY AND ANALYSIS PROTOCOL

We categorize failures as: (i) *surface* (syntax, missing imports, non-existent APIs), (ii) *format* (I/O mismatch, precision, delimiter mistakes), (iii) *semantic* (algorithmic logic, boundary conditions), (iv) *performance* (timeouts, non-termination). Labels are used only for qualitative diagnosis; they do not affect the primary metric.

## O  SECURITY AND SAFETY NOTES

We restrict network and filesystem access, enforce resource caps, and sanitize environment variables. Container images are minimized to reduce attack surface. Evaluation runs without network access; any package installation occurs during image build time.

## P  REPRODUCIBILITY CHECKLIST

- **Code**: training/eval harness, templates, manifests, logging of seeds and prompt IDs.

- **Data**: decontamination scripts, exclusion list (hash digests), dataset licenses and attributions.

- **Compute**: GPU/CPU details, container runtime, timeouts and limits.

- **Hyperparameters**: PPO clip $\varepsilon$, SBAM scale $\lambda$ (with $\gamma{=}1$), similarity temperature $\tau$, batch size, group size $G$.

- **Evaluation**: fixed templates, $T{=}0.2$, $M{=}20$ independent draws per prompt for pass@1 estimation, no best-of.

- **Uncertainty**: Wilson intervals, seeds, macro-averaging across prompts.

- **Safety**: sandboxing, resource caps, deterministic builds where possible.

## Q  LICENSE AND ATTRIBUTION NOTES

We respect dataset licenses and attributions. Where third-party benchmarks provide license terms (e.g., MultiPL-E), we follow them. Decontamination reduces inadvertent memorization of benchmark content; exclusion logs are maintained for auditability.

## R  NOTATION

| Symbol | Meaning |
| --- | --- |
| $x_j$ | Prompt (task) index $j$ in batch $\mathcal{B}$ |
| $y_{j,i}$ | $i$-th sampled response for $x_j$ |
| $G$ | Group size (responses per prompt) |
| $r_{j,i}$ | Binary reward $r(x_j, y_{j,i}) \in \{0, 1\}$ |
| $L_{j,i}$ | Sequence log-likelihood $\log \pi_{\text{old}}(y_{j,i}|x_j)$ |
| $h_j$ | Task embedding for $x_j$ (detached) |
| $w_{jk}$ | Similarity weight from $x_j$ to $x_k$ |
| $\mu_j^{V,\text{meta}}$ | Meta-mean for $V \in \{R, L\}$ |
| $\sigma_j^{V,\text{meta}}$ | Meta-std for $V \in \{R, L\}$ |
| $A_{j,i}^{\text{meta}}$ | Meta-normalized advantage proxy |
| $S_{j,i}$ | Surprise index $A_{j,i}^{\text{meta}} \cdot \widehat{L}_{j,i}^{\text{meta}}$ |
| $A_{j,i}^{\text{GMPO}}$ | Surprise-modulated advantage |
| $s_{j,i}(\theta)$ | Importance ratio $\pi_\theta/\pi_{\text{old}}$ |

## S    FURTHER PRACTICAL TIPS

**Numerical stability.**    Use denominator stabilizers $\delta > 0$; combine with gradient clipping. Threshold extreme z-scores for $\widehat{L}^{\text{meta}}$ if needed without altering sign.

**Template hygiene.**    Keep templates simple and consistent; avoid evaluation-specific hints. Prevent accidental inclusion of language-specific scaffolding in I/O-mode training.

**Diagnostics.**    Track failure categories and the distribution of $\widehat{L}^{\text{meta}}$ on success vs. failure to verify SBAM's emphasis mechanism qualitatively.

## T    RELATIONSHIP TO PRIOR PRACTICE

Our evaluation choices (I/O-style behavioral tasks for training; function-style tests for MultiPL-E) align with execution feedback Gehring et al. (2025) and multilingual evaluation Cassano et al. (2023); Athiwaratkun et al. (2023); Wang et al. (2023). We deliberately avoid best-of or reranking in the primary metric to measure single-draw performance under fixed decoding, reducing confounds from selection.

## U    THE USE OF LARGE LANGUAGE MODELS

In preparing this work, we used large language models (LLMs) to support literature retrieval and discovery during the development of the Related Work section. Specifically, LLMs were employed to identify relevant publications and summarize existing approaches in multilingual code generation benchmarks and reinforcement learning techniques for code LLMs. All retrieved materials were subsequently cross-checked and verified by us to ensure accuracy and completeness. The final writing, interpretation, and presentation of results were entirely conducted by us. Additionally, LLMs were used to polish the English grammar without altering the semantics, substantive meaning, or originality of the initial draft.

