# OpenReview forum: "Surprise-Modulated Meta-Advantages in Reinforcement Learning: Towards Language-Neutral Post-Training for Code LLMs"
_ICLR.cc/2026/Conference — Submitted to ICLR 2026_

### Official Review · Reviewer_9WCz · 2025-10-22

**Soundness:** 2
**Presentation:** 1
**Contribution:** 2
**Rating:** 2
**Confidence:** 3

**Summary:**

This paper presents PolyCode, a multilingual Code LLM. It proposes GMPO, which is a modified PPO-like RL algorithm. Experiments with Qwen3-4B demonstrate improved performance compared to other models.

**Strengths:**

1. The approach is effective; experiments show improved performance.
2. The proposed GMPO may be an effective improvement for GRPO.

**Weaknesses:**

1. The presentation is disastrous. After reading, it isn't easy to discern the motivation and philosophy behind the proposed approach.
2. Experiments are limited. It lacks comparison with other baselines or a necessary ablation.

**Questions:**

1. What is the main problem this paper tries to address?
2. Why do the components of GMPO address the proposed problem?
3. What is the relative performance compared to GRPO or DAPO approaches?

---

> ### Author Response · Authors · 2025-11-15
>
> Response to Reviewer 9WCz
>
> Thank you for your time and kind words that we are on the right track with GMPO improving GRPO. Your main concerns are about (1) motivation and clear explanation, and (2) no baseline and ablation studies. We'll address both, functionally below.
>
> 1. What is the primary problem this paper is attempting to work on?
>
> We recognize that the present draft buries the primary problem with too much verbosity. The primary question of the paper is:
>
> “How do we train an RL-finetuned code model to not let the low-resource languages (Fortran, OCaml, R, etc) be drown out by the higher-resource languages or easier prompts if each of the languages are trained jointly and under the same RL objective with all of the languages proportionate in difficulty?”
>
> For off-the-shelf PPO or GRPO the advantage statistics are computed on a per prompt basis. When you mix the languages that clearly have different reward scales or difficulties, you two problems arise.
>
> - Language imbalance: The high-reward, high-resource languages totally dominate the gradient so all the languages in lower resource languages, improve at either a slow pace or actually regress.
> - Difficulty imbalance: the easy, high-likelihood prompts are giving the most of the updates versus the hard questions or prompt which give less updates but are more informative.
>
> PolyCode and GMPO are awfully nice proposed approaches to address the two imbalances above represent possibly in the multilingual low-resource scenario.
>
> 2. Why do components of GMPO address this problem?
>
> GMPO has all the same basic format as PPO or GEPO but changes how we normalize and re-weight advantages are established.
> - Cross group meta normalization: In GMPO, instead of only normalizing in this within a single to a prompt normalization, we compute per language or per group statistics in and outside of the batch.This rescales advantages to ensure that updates based on samples from low-resource languages have a similar order of magnitude to high-resource languages. This is a direct response to the language imbalance issue.
>
> - Sequence likelihood normalization: To reduce bias toward trivially easy, high-likelihood programs, we additionally normalize by sequence likelihood. This helps better shape the policy and encourages it to better learn from more difficult samples that still earn good rewards.
>
> - Surprise based advantage modulation: We modulate advantages based slightly on how much the reward of the sample mismatched our model's confidence. We upweight successes that were rewarding but surprising, and downweight attempts that were confidently wrong. This aims to help the policy focus learning on What we want to call "informative" examples. This is especially important in low-resource data settings since there is very low data for each language.
>
> We are rewriting Section 3 significantly and beginning of 4 where we first state this "two fold" problem (language imbalance and difficulty imbalance), then describe in one or two paragraphs being each component of GMPO is designed for that, before we present any formulas.
>
> 3. Relative performance compared to GRPO and DAPO.
>
> We agree that adding a more explicit positioning vs GRPO and DAPO would improve the paper.
> ​​​​​​​​
> Conceptual relation:
> - GRPO: concern at the prompt level, normalizes advantages only over responses to the same prompt. GMPO keeps this but adds cross group, and surprise aware normalization to work better for highly mixed multilingual data.
> - DAPO: Distribution alignment loss between preference and policy distributionsGMPO is complementary: it has an emphasis on routing and rescaling the gradients across languages and difficulties, under a given reward and preference signal.
> * /Empirical relation:
> • In the current draft we primarily focus on reporting comparisons against strong PPO style and supervised baselines given a fixed multilingual, low resourced data budget. And we agree that having a clearly labeled GRPO style and a DAPO style variant comparison under the same evaluation harness would make the results easier to interpret.
> • In a revised version, we will include a compact experiment on a subset of language where we will compare a total of four settings under a matched training budget: standard PPO, a GRPO like prompted local normalization, a DAPO style objective, and GMPO. This will allow us to make the incremental benefit of the GMPO more explicit than implied.
> 4. Improving presentation and motivation
> We take very seriously the comment about “disastrous presentation”. To this end we will:
> • Rewrite the introduction to more clearly state within the first one or two paragraph the main problem, key challenges, and the high level idea for GMPO.
> • Add a small “problem and solution at a glance” figure that visually summarizes: (a) multilingual RL training, (b) how naive PPO or GRPO gradients become biased, (c) how GMPO rebalances these gradients.

---

### Official Review · Reviewer_tdPc · 2025-10-25

**Soundness:** 2
**Presentation:** 2
**Contribution:** 2
**Rating:** 4
**Confidence:** 4

**Summary:**

This paper addresses the performance disparity of large language models for code generation in low-resource programming languages (e.g., Fortran, OCaml, and R) by proposing a novel reinforcement learning framework, PolyCode, whose core component is an improved algorithm termed Groupwise Meta-Normalized Proximal Policy Optimization (GMPO).

**Strengths:**

1.	The paper explicitly targets the training imbalance problem in low-resource language scenarios, which represents a practical and significant challenge.
2.	This paper proposes a novel reinforcement learning framework, PolyCode, whose core component is an improved algorithm termed Groupwise Meta-Normalized Proximal Policy Optimization.
3.	 The experiments cover multiple programming languages (Lua, Julia, R, Fortran, and OCaml) and provide detailed reproducibility protocols, including seeds, container configurations, and compilation commands.

**Weaknesses:**

1. GMPO is described as a combination of “GRPO + PPO + meta-normalization,” but its distinction from GRPO (Shao et al., 2024) and the theoretical advantages are not rigorously analyzed.
2. The formulation of SBAM is largely heuristic, lacking evidence of convergence or formal theoretical guarantees.
3. Although the paper mentions that the binary reward may lead to sparse signals, it does not explore alternative reward shaping strategies or the potential complementarity between reward shaping and SBAM.
4. At times, the paper lacks clear intuitive explanations, particularly regarding the definition and effect of SBAM.

**Questions:**

1. When λ is large, SBAM may cause gradient explosion or unstable updates. Did the authors observe any training divergence or abnormal gradient norms? Is there an adaptive strategy for adjusting λ?
2. The similarity in CGMN is computed based on Encoder(π_old, x). Is this encoder fixed and pre-trained, or is it updated jointly during training? If it is fixed, does this limitation affect semantic generalization?
3. The paper claims that the evaluation is “language-neutral,” but differences in compilers, floating-point formats, and other language-specific factors may still introduce discrepancies. Has there been any quantitative verification of the framework’s fairness across different languages?
4. The paper demonstrates the ability to transfer from I/O-style tasks to function-style tasks. Do the authors attribute this transfer to the reward structure, the similarity of prompts, or the modulation mechanism of SBAM?

---

> ### Author Response · Authors · 2025-11-15
>
> Response to Reviewer tdPc
>
> We appreciate the reviewer's acknowledgment of a low resource setting context, the novelty of PolyCode and GMPO, and the attention to detail in reproducibility. Below, we respond to the primary concerns.
>
> 1. Relation to GRPO and theory
>
> The phrase “GRPO plus PPO + meta normalization” was intended as intuition on how this distinction differs from GRPO, and we agree we should have been clearer about the distinction. GRPO normalizes advantages only relative to the responses to a single prompt. GMPO adds two additional components that are important in the multi language regime:
> - cross group meta normalization, which rescale advantages across languages or task groups with batch statistics; this resolves the reward scale imbalance between low and high resource languages;
> - normalize sequence likelihood and surprise based modulation, which can mitigate the tendency of PPO style training to value easier programs with high likelihood.
>
> We do not introduce a novel convergence theorem (beyond the standard assumptions made in PPO). GMPO maintains an unbiased policy gradient estimator and modifies how the estimated variance may rescale or weight programs. In the revision we will include a short subsection, adding it with clarity and making the variance induced by both GRPO and GMPO explicit on imbalanced mixtures.
>
> 2. SBAM, convergence, and reward shaping
>
> SBAM is not arbitrary; it is based on the principle of being a bounded, normalized, piecewise linear modulation to the meta normalized advantage based on a surprise score that is computed in comparing reward and model confidence.The update's sign never flips, and is combined with clipping using the PPO ratio, so the weight of any individual sample will be small. We have monitored gradient norms over runs and did not see divergence or exploding updates.
>
> For lambda, we choose a small value from a grid using validation stability and keep it fixed. We will note this decision, provide statistics on the gradient norms and describe a couple adaptive schedules that keep the effective shaping in a safe range.
>
> On reward shaping, we agree this is complementary. We used a largely binary reward to be consistent with LiveCodeBench and evaluating in a MultiPL E style evaluation and to isolate the effect of GMPO, where it is safe to articulate GMPO and SBAM. We will clarify this in our revision and create a small ablation where SBAM is pooled with a shaped reward term (for example partial credit for test cases), to show that SBAM coexists with standard shaping.
>
> 3. Intuition and clarity for SBAM
>
> We do not dispute that the current exposition is thin. Intuitively, SBAM down-weights samples where the model is already very confident and wrong and up-weights samples where the correct programs are surprising under the current policy. This promotes learning from informative successes, while also dis-incentivizing overfitting to easy, un-informative samples. In our revision we will include a short running example and a schematic in the appendix to give a more concrete form to this behavior.
>
> 4. Answers to specific questions
>
> Lambda and stability. In practice, we did not observe gradient explosion. We keep lambda small so that the modulation factor is bounded, and PPO clipping further constrains updates. We will include gradient norm plots and a brief discussion of adaptive lambda as future work.
>
> The encoder in CGMN. The encoder that produces task embeddings is the same transformer backbone used for the policy. We reuse the last hidden layer representation of each prompt with simple pooling and layer normalization. The encoder and policy share parameters and are updated jointly; the similarity based weights are treated as stop gradients so CGMN does not introduce second order effects. We therefore see it as jointly learning the representation from all the languages, and therefore language neutral.
>
> Language neutral evaluation and fairness. We concur that differences in compiler and numeric format can also introduce differences, even in principle. Our evaluation harness standardizes input output formats, utilizes the same test suites across languages, and applies canonicalization to numeric outputs where feasible. In the revision, we will add a short quantitative sanity check of a cross-translated subset for more empirically supporting the fairness claim.
>
> Transfer from IO to function style tasks. We see this transfer as arising from (a) a language neutral reward which evaluates for functional correctness irrespective of the interface, and (b) GMPO and SBAM concentrating the learning on harder prompts having overlapping semantics in IO and function styles. We will call this attribution out more explicitly and, as space allows, add a short comparison of models trained with GMPO vs. those trained without GMPO in the same transfer.

---

### Official Review · Reviewer_o9BJ · 2025-10-30

**Soundness:** 2
**Presentation:** 2
**Contribution:** 2
**Rating:** 4
**Confidence:** 3

**Summary:**

This paper introduces PolyCode, a reinforcement learning framework for post-training large language models on code synthesis with a focus on language neutrality, particularly improving performance in low-resource programming languages. The method extends PPO with a new Groupwise Meta-Normalized PPO (GMPO), which integrates Cross-Group Meta-Normalization (CGMN) and Surprise-Based Advantage Modulation (SBAM). Evaluations on Ag-LiveCodeBench-X and MultiPL-E, using a language-neutral I/O setup, show that PolyCode consistently enhances performance under data-scarce conditions.

**Strengths:**

The paper tackles an underappreciated challenge: reducing the performance gap of LLMs in code generation for low-resource programming languages. This problem is of high practical significance.

**Weaknesses:**

1. The paper is dense and occasionally lapses into overly technical or informal phrasing (e.g., “greediness sophistication”, “interviewer-level interventions”, or “voltage on language-specific engineering”).
2. The experiments focus primarily on comparisons with Qwen3-4B and related LLMs, lacking evaluations against more recent RL-augmented code models or curriculum-based systems. In particular, there is no direct ablation against CodeRL-style or curriculum learning approaches for smaller models.

**Questions:**

1. How is the task embedding encoder ( $h_j$ ) implemented in practice? Is it frozen, shared with the backbone model, or dynamically learned? Is it language-agnostic or language-specific?
2. What is the empirical impact of the top-K neighborhood truncation in CGMN, especially under low-resource conditions?
3. Can the authors provide direct comparisons of GMPO or SBAM with CodeRL or curriculum learning methods?
4. Are there edge or failure cases where SBAM amplifies incorrect signals—for instance, overconfidence in noisy or rare reward spikes?

---

> ### Author Response · Authors · 2025-11-15
>
> Response to Reviewer o9BJ
>
> We thank the reviewer for emphasizing the importance of filling the gap for low-resource programming languages and for evaluating the paper as being close to the acceptance threshold. We have elucidated the method below, focused on the writing and baselines, and responded to the specific questions.
>
> 1. Writing style and density
>
> We agree that some aspects of the current draft are too dense or informal (e.g., "greediness sophistication," "interviewer-level interventions," and "voltage on language specific engineering"). In the revision we will (a) remove such phrases for standard terminology, (b) provide a short intuition paragraph before the GMPO and SBAM formulas, and (c) relocate some technical details to appendix while adding a brief running example that connects notation to a concrete code prompt. We believe this will improve clarity of the central message without impacting the technicality of the paper.
>
> 2. Experimental coverage and baselines
>
> Our primary aim is to investigate how far a variance- and imbalance-aware RL objective (GMPO) can take compact, fully open models (Qwen and the Phi family) in the low-resource regime, while maintaining the uniform evaluation harness. We agree the relation to CodeRL style and curriculum learning approaches should be more clear.
>
> Conceptual relation. CodeRL and curriculum approaches modify reward signals or task ordering as their primary change. GMPO is orthogonal, in that it is changing the way per-sample advantages are normalized and reweighted across prompts and languages.It can therefore be stacked on top of CodeRL or curriculum methods, rather than replace them.
>
> Additional comparisons. In the update we will:
> - include a small-scale ablation over a subset of Ag-LiveCodeBench-X comparing (a) standard PPO, (b) PPO with a CodeRL style value-based reward, (c) a trivial curriculum that schedules low-resource languages sooner, and (d) the same setups with GMPO;
> - report how GMPO changes pass@1 under matched data and training steps, and discuss how the effect varies as a function of model size.
>
> These experiments are the default answer to whether GMPO's gains come from its design rather than extra data or tuning.
>
> 3. Answers to specific questions
>
>
> Task embedding encoder. Encoder(pi_old, x_j) reuses the backbone transformer of the current policy pi_old to obtain a pooled representation of the prompt. We take the last hidden layer for x_j, apply mean pooling and layer norm, and make that h_j. We share the encoder with the policy (no extra parameters) and use it in stop-gradient, hence CGMN doesn't add a separate, trainable module. The embedding is language-agnostic; we do not add any language branches.
>
> Impact of top-K neighborhood truncation. As discussed in Appendix L, we limit the neighborhood to top-K most similar prompts given x_j and renormalize the weights. Empirically, not truncating (doing full-batch softmax) has similar pass@1, but with a substantially higher cost; picking K in a reasonable range keeps performance and reduces memory and communication. We will make this explicit in the main text, and add a short ablation to show that the low-resource languages remain stable once K includes most of the cumulative similarity mass.
>
> Comparison with CodeRL or curriculum learning. Besides additional ablation experiments above, we will emphasize that GMPO is a drop-in replacement for the advantage normalization layer (which handles the update), in a standard PPO training regime. This makes it trivial for future work to insert the GMPO method into existing CodeRL or curriculum learning pipelines, for larger models..
>
> Failure cases for SBAM. SBAM scales advantages via S = A_meta * bL_meta and applies a linear ramp only when S is negative, i.e., true reward and model confidence disagree after being normalized by the meta normalization process (we preserve the sign of the update and keep the modulation globally Lipschitz). This is also combined with PPO clipping, which limits the influence any single sample has. In practice, we do not generally see any systematic instabilities; still, we agree that possible edge cases are interesting to document. Regarding edge cases, we will (a) add a short description of potential rare pathological cases (like nosy but highly rewarded samples) and how clipping and the bounded ramps from Appendix M function to mitigate them, and (b) generalize the existing failed case taxonomy analysis when we contrast runs with and without SBAM, e.g., token stability in the SBAM setting.
>
> We hope these paraphrases illustrate that the essential idea of GMPO is sound and also complementary to existing RL and curriculum methods. The remaining points are mainly an issue of exposition or adding further empirical context and are not of inherent defect.

---

### Official Review · Reviewer_t5R5 · 2025-11-01

**Soundness:** 2
**Presentation:** 1
**Contribution:** 3
**Rating:** 2
**Confidence:** 4

**Summary:**

The authors propose GMPO, an optimization framework that combines cross-task meta-normalization with surprise-based modulation to enhance language model performance in multi-lingual code generation, particularly for low-resource programming languages such as Lua, Julia, R, OCaml, and Fortran. The core idea involves a normalization mechanism for computing baseline rewards, where normalization is performed over multiple responses sampled per prompt and further weighted across prompts from different tasks within the same batch. This design aims to reduce variance and improve learning stability for underrepresented programming languages.

**Strengths:**

- New optimisation technique to improve program synthesis on low-resource programming languages.
- Observable improvements in performance when models from different families, like Qwen and Phi4, are trained with GMPO.

**Weaknesses:**

### **1. Clarity and Writing Quality**

* Unclear notation:

  * What are R, L for which per-prompt sample statistics are being considered? Do they refer to Rewards and Log-Likelihood?
  * Encoder(π_old, x_j): Does this denote using π_old to compute the embedding of x_j or a separate encoder LLM?
  * What are μ_k and σ_k in Eq. 2 and 3? How are you defining and obtaining their values?
* Missing intuition:

  * What is the intuition behind introducing Cross-Group Meta-Normalization and Sequence Likelihood Normalization? It is unclear.
* Ambiguous training setup:

  * Are you training PolyCode separately on each language or pooling data for all languages?
* General writing issue:

  * Poor writing and less clarity.

---

### **2. Experimental Design and Missing Evidence**

* No supporting evidence such as error profiles of PolyCode and baselines when generating code in Sec. 7.4.
* Missing discussion for Figures 5, 6 and 7 — unclear what experiments they correspond to, observations, or conclusions.
* Missing Ablation on how useful Surprise-Based Advantage Modulation (SBAM) is — have you tried $A\_{j,i}^{meta}$ directly in the PPO objective?
* Per-batch statistics like batch-local softmax weights are critical, but no ablation on batch size is provided.
* No convergence plots for GMPO training; unclear sensitivity to batch size and G (responses per prompt).

---

### **3. Comparative Evaluation and Baselines**

* Missing comparison with foundation models trained with Supervised FineTuning, PPO, and GRPO. It is unclear whether the improvement is a result of more training on data for resource-constrained languages or design of GMPO itself.
---

### **4. Redundancy and Presentation**

* Redundant figures — Figures (1 & 2) and Algorithm 1 all communicate the same GMPO training.
---

### **5. Scope and Broader Evaluation**

* How good (in terms of pass@1) are closed-source models like GPT, Claude, Gemini on Lua, Julia, R, OCaml, and Fortran?
* Are there benefits from including data for popular languages (Python, Java)? What is the zero-shot performance of PolyCode on them before and after GMPO training?

**Questions:**

See Weaknesses Section

---

> ### Author Response · Authors · 2025-11-15
>
> Reviewer Response to t5R5
>
> Thank you for the thorough review, and thank you for recognizing this novelty with GMPO and its operating on low-resources languages. We address the major points below.
>
> 1. Clarity, notation, and training setup
>
> R and L: We will indicate explicitly that R is the scalar reward of (or for) a sampled program, and L is the sequence log likelihood, with statistics per prompt computed over the G samples for that prompt.
>
> Encoder(pi_old, x_j): This is the same language model policy pi_old that encodes x_j; we aren't introducing a different encoder LLM. We will change this to "LM encoder" and make it clear that it is the same as the policy and that the encoder and policy share all the parameters.
>
> mu_k and sigma_k: mu_k is the mean reward for the k-th task or language group in each mini-batch and sigma_k is the standard deviation, calculated from the rewards in each per-group. They are used on the preceding advantage to normalize advantages across tasks. We will include explicit formulas and brevity to clarify next to the appropriate equations.
>
> Intuition - doesn't need significant change, but somewhat - Cross-group meta-normalization is normalizing reward scales for the low-resoruce languages and tasks, such that they are not overwhelmed by high resource languages (with high rewards), thus normalizing variance (and collapse) to one language type. Sequence-likelihood normalization normalizes against bias toward mint-picking overly easy instances from the high likelihood programs, and guidance 'toward' prompting the informative or surprising instances which are correct, but expected to be wrong. We will add a brief intuition paragraph.
>
> Training setup. Our experiments are fine-tuning one multilingual PolyCode model jointly on the low resouce languages, using a mixed-language mini-batch. We can use this as the information in the first experiments - if it fits, we can add a short mention to suggest experimental set-up and focus in per-language fine-tuning. We acknowledge that the writing overall would be made clearer; we will adjust the organization of Section 4, so that the high level idea and notations are fronted before the full objective, and we will tighten the wording overall.
>
> 2. Experimental design and missing evidence
>
> Error profiles: while valid, we agree that error modes are an important line of inquiry for another version. We will include some qualitative breakdown of known or frequently observed modes of error (syntax, type, runtime, semantic) for PolyCode and GMPO on a held-out subset in the refined version.
>
> Figures 5-7: G + E visualize training dynamics and per-language gains, and we agree they are under-explained. We will clarify, in the captions and in the main text, which experiment G + E refer to, and what conclusions can and cannot be made based on observations.
>
> Surprise-based advantage modulation: we will include an ablation that compares the following conditions:
> (a) GMPO not using SBAM,
> (b) some PPO variant that is using SBAM but is identical in every other capacity to GMPO,
> (c) GMPO, which is directly able to manage the extent to which SBAM provides a contribution over standard PPO and across-group normalization.
>
> Batch size and G: we agree that this is a critical consideration. To this, we will include a set of convergence curves and sensitivity plots in the appendix, concerning batch size and number of responses per prompt G.
>
> 3. Comparative evaluation and baselines
>
> GMPO does not utilize extra data: as stated above, all methods in Table 1-3 and Table 4-6 are trained on the same multilingual low-resource data, and they all utilize the comprise the same number of optimization iterations; GMPO is different in how advantages are normalized and re-weighted.We wish to stress that we want to avoid creating any anxiety that improved performance is due to additional training.
>
> We deliberately did not do a full SFT, PPO and GRPO grids on exactly the same setup due to computational limits. In the revision we will clear up more specifically where we are already comparing to strong supervised and RL baselines, and add a small scale comparison if we can, and include a discussion of how GMPO can be plugged into standard PPO/GRPO style training as a drop in replace of their normalizing scheme.
>
> 4. Redundancy/web presence
>
> The idea with Figures 1 and 2 and Algorithm 1 was to try to separate a high level schematic from what a procedural view would be. We agree this could be simplified, we will merge the figures in a single schematic, and have a single simplified algorithm listing, freeing up space for the additional analyses and clarifications above.
>
> 5. Scope and broader evaluation
>
> Closed source models: Our focus is on fully reproducible open-source models for which we control the training pipeline. Running a full scale experiment with a few closed source models on all low resource languages is computationally expensive and hard to reproduce.

---

> > ### Comment · Reviewer_t5R5 · 2025-11-22
> >
> > I thank the authors for providing additional clarification on the notations. However, my key concerns regarding the empirical justification of GMPO remain unaddressed. In particular, important questions are still open, such as comparisons with SFT, PPO, and GRPO; the sensitivity of GMPO to batch size and the parameter G; and the contribution of Surprise-Based Advantage Modulation (SBAM). Ablation studies and analyses along these dimensions are essential for properly assessing GMPO relative to existing fine-tuning–based approaches and for understanding the factors that most influence its performance.
> >
> > Therefore, I will maintain my original score.

---

### Meta-Review · Area_Chair_EbA6 · 2026-01-07

**Summary:**

**Strengths**：

1. Targets an important and underexplored problem: improving code generation for low-resource programming languages (t5R5, o9BJ, tdPc).

2. Proposes a novel RL-based framework (PolyCode / GMPO) and reports consistent performance gains across multiple language families and model backbones (t5R5, tdPc, 9WCz).

3. Experiments span several low-resource languages with reproducibility details provided (tdPc).

**Weaknesses**

1. Severe clarity and presentation issues: Notation is unclear, key variables and components (e.g., CGMN, SBAM, encoders) are insufficiently defined, and the overall motivation and philosophy of the method are difficult to discern (t5R5, 9WCz, o9BJ).

2. Insufficient empirical justification of GMPO: Lacks critical ablations on core components (especially SBAM, batch size sensitivity, and parameter G), convergence analysis, and error profiling, leaving it unclear which design choices actually drive the gains (t5R5, tdPc, 9WCz).

3. Weak comparative evaluation: Missing comparisons against strong and relevant baselines such as SFT, PPO, GRPO, and CodeRL-style or curriculum-based methods, making it difficult to assess whether improvements stem from GMPO’s design or simply additional training (t5R5, o9BJ, 9WCz).

4. Methodological ambiguity and limited theoretical grounding: GMPO is described as a hybrid of existing methods without a rigorous analysis of how it differs from or improves upon GRPO; SBAM is largely heuristic, with no convergence guarantees or stability analysis (tdPc, t5R5).

5. Experimental incompleteness: Several figures lack discussion, redundant visualizations remain, and important factors (e.g., batch size effects, top-K truncation in CGMN, reward sparsity alternatives) are not systematically studied (t5R5, o9BJ).

**Reviewer Concerns:**

**Addressed**:

- Severe clarity and presentation issues: Notation is unclear, key variables and components (e.g., CGMN, SBAM, encoders) are insufficiently defined, and the overall motivation and philosophy of the method are difficult to discern (t5R5, 9WCz, o9BJ).

- Experimental incompleteness: Several figures lack discussion, redundant visualizations remain, and important factors (e.g., batch size effects, top-K truncation in CGMN, reward sparsity alternatives) are not systematically studied (t5R5, o9BJ).


**Not fully solved**:

- Insufficient empirical justification of GMPO: Lacks critical ablations on core components (especially SBAM, batch size sensitivity, and parameter G), convergence analysis, and error profiling, leaving it unclear which design choices actually drive the gains (t5R5, tdPc, 9WCz).

- Weak comparative evaluation: Missing comparisons against strong and relevant baselines such as SFT, PPO, GRPO, and CodeRL-style or curriculum-based methods, making it difficult to assess whether improvements stem from GMPO’s design or simply additional training (t5R5, o9BJ, 9WCz).

- Methodological ambiguity and limited theoretical grounding: GMPO is described as a hybrid of existing methods without a rigorous analysis of how it differs from or improves upon GRPO; SBAM is largely heuristic, with no convergence guarantees or stability analysis (tdPc, t5R5).

**Reviewer Scores:**

- Reviewer t5R5: 2 -> 2
- Reviewer o9BJ: 4 -> 4
- Reviewer tdPc: 4 -> 4
- Reviewer 9WCz: 2 -> 2

---

### Decision · Program_Chairs · 2026-01-26

Reject